# Anakinra for the Treatment of Hemophagocytic Lymphohistiocytosis: 21 Cases

**DOI:** 10.3390/jcm11195799

**Published:** 2022-09-30

**Authors:** Clara Baverez, Maximilien Grall, Mathieu Gerfaud-Valentin, Sarah De Gail, Alexandre Belot, Thomas Perpoint, Emmanuelle Weber, Quitterie Reynaud, Pascal Sève, Yvan Jamilloux

**Affiliations:** 1Service de Médecine Interne, Hôpital de la Croix Rousse, Hospices Civils de Lyon, Université Claude Bernard-Lyon 1, 69008 Lyon, France; 2Service de Médecine Intensive et Réanimation, Hôpital Hospitalo—Universitaire de Rouen, Université de Rouen, 76000 Rouen, France; 3Service de Médecine Interne, Hôpital Hospitalo—Universitaire de Rouen, Université de Rouen, 76000 Rouen, France; 4Service de Rhumato-Néphro-Dermatologie Pédiatrique, Hôpital Femme Mère Enfants, Hospices Civils de Lyon, Université Claude Bernard-Lyon 1, 69008 Lyon, France; 5Lyon Immunopathology FEderation (LIFE), Université Claude Bernard-Lyon 1, 69100 Lyon, France; 6Service de Maladies Infectieuses et Tropicales, Hôpital de la Croix Rousse, Hospices Civils de Lyon, Université Claude Bernard-Lyon 1, 69008 Lyon, France; 7Service de Médecine Interne, Centre Hospitalier Lyon Sud, Hospices Civils de Lyon, Université Claude Bernard-Lyon 1, 69008 Lyon, France; 8Research on Healthcare Performance (RESHAPE), INSERM U1290, University Claude Bernard Lyon 1, 69100 Lyon, France

**Keywords:** anakinra, interleukin-1, hemophagocytic lymphohistiocytosis, adult onset Still’s disease, immunotherapy, cytokines, inflammatory mediators

## Abstract

Hemophagocytic lymphohistiocytosis (HLH) is a life-threatening cytokine storm syndrome. There are no definitive guidelines for the management of secondary HLH (sHLH). A recent report by a National Health Service (NHS) clinical panel has recommended anakinra as a treatment option. We aimed to analyse the efficacy and safety of anakinra for the treatment of all-cause sHLH. We conducted a multicentric retrospective study in two French University hospitals and included all patients who had a diagnosis of sHLH and who received anakinra. Among 21 patients (median age, 45 years), 13 were men. Anakinra was used as first-line therapy in 10 patients, and as monotherapy in 5 patients. We found that anakinra was effective in 19/21 patients (90.5%), with fever resolution in 19 patients (90.5%) within a median of 1.0 day (1, 2). At the Day 7 assessment, the mean CRP concentration decreased significantly (*p* < 0.001), as did the mean ferritin (*p* = 0.011). Anakinra was generally safe and well tolerated and was discontinued for side effects in only three patients (14.3%). Anakinra is an efficient and safe treatment to control sHLH of various causes. These data, together with the recent report of the NHS panel, call for the rapid conduct of prospective randomized clinical trials.

## 1. Introduction

Hemophagocytic lymphohistiocytosis (HLH) is a life-threatening cytokine storm syndrome characterized by non-remitting fever, hepatosplenomegaly and cytopenia. HLH encompasses several conditions, with primary HLH (pHLH) being genetically determined, while secondary HLH (sHLH) may complicate infections, malignancies, immunodeficiencies or rheumatic diseases [1]. In the latter case, it most often occurs in the context of Still’s disease or systemic lupus erythematosus, and is termed macrophage activation syndrome. HLH results from an uncontrolled hyperinflammatory response, characterized by hyperferritinemia, elevated levels of proinflammatory cytokines (including IFN-γ, IL-1β, IL-6, IL-12, IL-18 and TNF) and hemophagocytosis [2].

Unlike pHLH [3], there are no single guidelines for the management of sHLH. The first reason is probably the lack of recognition of sHLH as a per se entity requiring specific treatment (i.e., as an aggravating condition), with the attention of physicians being focused on treating the underlying cause. The second reason is the heterogeneity (and sometimes co-existence) of the underlying aetiologies, some of which even make the use of immunosuppressive drugs counter-intuitive (infection, lymphoma). Finally, the diagnosis of sHLH can sometimes be missed or neglected due to the absence of sensitive and specific markers and validated diagnostic scores for all these situations.

In addition to treatment of the underlying condition, management of sHLH requires rapid suppression of the hyperinflammatory response. Steroids are usually the mainstay of initial immunosuppression. Etoposide is considered for refractory sHLH or in case of organ failure [4]. However, etoposide yields haematological toxicity and caution is advised in severe renal and liver impairment.

Despite the low-quality of evidence, the recombinant IL-1 receptor antagonist (IL-1Ra), anakinra, has recently been recommended by some authors [4,5] and an NHS clinical panel [6] as a second-line treatment for sHLH, after corticosteroid failure. This recommendation was based on single-centre retrospective case series of paediatric [7,8,9] and adult patients [10,11], and on a post hoc analysis of a randomized controlled trial in sepsis that suggested a benefit of anakinra in selected patients [12]. The recent SAVE-MORE study confirmed the efficacy and safety of anakinra in patients with features of SARS-CoV2-induced sHLH [13], providing an encouraging and reassuring signal in sHLH, even due to infection.

## 2. Materials and Methods

We conducted a multicentric retrospective study in two University hospitals (Lyon and Rouen, France) to evaluate the efficacy and safety of anakinra in sHLH of any etiology. We searched our local database from January 2012 to April 2022 with the terms “HLH”, “macrophage activation syndrome” and “anakinra”, and performed a call for observation. Patients who had sHLH defined by an H score > 169 (169 being the best cut-off value for H Score in the study of development and validation of the H score [14]), a clinician-confirmed diagnosis and who received anakinra in this setting were included. 

Clinical, demographic and laboratory data were analysed and descriptive statistics computed for each variable. Categorical variables are presented as numbers and proportions, and continuous variables as medians with interquartile ranges [IQR]. Groups were compared using the Fisher’s exact test (for categorical variables) or the Wilcoxon test (for continuous variables); *p* < 0.05 was considered significant. We used Microsoft Excel version 2019 (Redmond, WA, USA) for all analyses.

All patient data were collected retrospectively. This study was conducted in compliance with good clinical practices and the principles of the Declaration of Helsinki. In accordance with local laws, formal approval from an ethics committee was not required for this type of retrospective study.

## 3. Results

Among 21 patients, with a median age of 45 years [IQR, 33–58], 8 were women and 13 were men. There were three children. The median time from onset of sHLH symptoms to sHLH diagnosis was 12 days [IQR, 8–15]. At least one sHLH-triggering condition was identified in all cases: six rheumatic diseases (three Still’s disease and three systemic-onset juvenile idiopathic arthritis), five malignancies (two lymphomas, two myelodysplastic syndromes and one urothelial carcinoma) and twelve infections (four tuberculosis, three bacterial infections, two fungal infections and three viral infections).

sHLH was the inaugural manifestation of the predisposing disease in 15 patients (71%). The main symptoms were fever (*n* = 21), hepatosplenomegaly (*n* = 15), lymphadenopathy (*n* = 12) and serositis (*n* = 9). The median ferritin level was 13,835 ng/mL and the median H score was 244, corresponding to a median HLH probability of 99.1%. A bone marrow aspiration was performed in all patients, and retrieved hemophagocytosis in 16 (76%) cases (Table 1).

Anakinra was used as first-line treatment for sHLH in 10 patients. In six patients (five with infection, one with Hodgkin’s lymphoma), anakinra was administered as monotherapy without corticosteroids. None of these six patients required a new line of treatment for sHLH; only one patient was switched to oral corticosteroids to facilitate drug intake. In four patients (two with rheumatic disease, one with Hodgkin’s lymphoma and one with infection), anakinra was combined with corticosteroids. None of these patients warranted a new line of treatment. Anakinra was used as a second-line treatment after failure of corticosteroid therapy alone in seven patients (two with infection, two with myelodysplastic syndrome, two with rheumatic disease and one with graft rejection,). None of these seven patients required a new line of treatment for sHLH. Anakinra was prescribed as a third-line treatment in two cases (one with rheumatic disease, one with infection), after failure of polyvalent immunoglobulin and then steroid therapy alone. In patient 15, who had juvenile idiopathic arthritis and primary parvovirus B19 infection, a fourth line of treatment with ciclosporine was required after failure of anakinra. Anakinra was used as a fourth-line therapy in patient 3 who presented sHLH in the context of Crohn’s disease treated with azathioprine, after the use of corticosteroids, polyvalent immunoglobulin and etoposide. Anakinra was not effective in this patient. Finally, anakinra was used as fifth-line therapy in patient 9 with Still’s disease, without the need for a new line of therapy (Table 2, Appendix A).

The drug was always delivered subcutaneously, at dosages of 100 mg/day (*n* = 14), 200 mg/day (*n* = 5), or 2–5 mg/kg/day in children (*n* = 2). 

According to the clinician, anakinra was effective in 19 patients (90.5%). In three cases, the dose of anakinra had to be doubled to achieve remission. Fever resolved in 19 patients (90.5%) within a median of 1.0 day [IQR, 1–2]. At the Day 7 assessment, the mean CRP concentration decreased significantly (*p* < 0.001), as did the mean ferritin (*p* = 0.011), but there was no statistical difference in cytopenia (Table 3). 

Fifteen patients achieved a CRP < 10 mg/L, within a mean delay of 12 days [IQR, 7.5–32.0]. The average duration of hospitalization was 16 days [IQR, 10–32] after starting anakinra. The median duration of treatment with anakinra was 21 days [IQR, 13–19], with shorter durations for malignancies or infections than for Still’s disease (median, 17 vs. 195 days, *p* = 0.016). At last follow-up, two patients with rheumatic diseases were still receiving anakinra after two and five years, respectively. 

Anakinra was generally well tolerated; five patients (23.8%) had side effects: two injection-site pain, one erysipelas, one pneumonia, and one allergic reaction. Treatment discontinuation was required in three cases. 

During a median follow-up of 10 months [IQR, 4.0–22.0], three patients died: two deaths were due to the underlying hematologic malignancies and one to an uncontrolled sHLH (Patient 3). Four patients experienced a recurrence of sHLH. All cases were controlled either by increasing the dosage of anakinra (*n* = 4), or by combining CS (*n* = 1), CSA (*n* = 1), or azacytidine (*n* = 1; Table 3, Appendix A).

## 4. Discussion

In this study, the use of anakinra in 21 patients with sHLH of various aetiologies was associated with a favourable outcome in 19 patients (90.5%). The resolution of symptoms was rapid and biological inflammation improved within the first week. However, cytopenia took longer to improve (with or without underlying hemopathy). Apart from injection-site pain, side effects occurred in <15% of cases in immunocompromised subjects. Anakinra was used without corticosteroids in five patients, which allowed us to carry out the diagnostic investigations without fear of negative histological samples.

Unlike some other HLH-specific treatments, anakinra seems to have a satisfactory safety profile and could be preferred to etoposide, at least as a therapeutic test during the first 24–72 h, due to a lower septic risk and the absence of haematological toxicity, or to ciclosporine, due to its risk of neurological toxicity [4]. In cases of EBV-associated sHLH or multi-visceral failure, etoposide remains a first-line option. However, it is possible that anakinra can be used in addition to etoposide, without significantly increasing the infectious risk, to achieve rapid control of a deleterious cytokine storm. Moreover, its short duration of action makes it more manageable than other specific treatments or biotherapies such as IL-6 inhibitors. 

Limitations of this study include the small number of patients and its retrospective nature in relation to the low prevalence of HLH. The small sample size prevented deeper analyses, such as the search of predictors of response. Furthermore, judging the efficacy of anakinra in sHLH is made more difficult by the fact that it is a disease that is itself associated with many different pathological conditions that warrant specific treatment, and as anakinra is not currently approved for use in this indication, the administration regimens are left to the discretion of the clinician. However, our data are consistent with the efficacy of anakinra and add original cases of anakinra use in real life to the literature, which will inform future systematic reviews and are useful until a prospective work can provide precise answers.

## 5. Conclusions

We report encouraging data on the efficacy and safety of anakinra in the control of sHLH of various causes. Prospective studies are needed to fully confirm this efficacy and to precisely define the place of this drug in the management of sHLH.

## Figures and Tables

**Table 1 jcm-11-05799-t001:** Clinical and laboratory findings in 21 patients with secondary HLH.

	Number of Patients (%) or Median [IQR]
Gender (male)	13 (62%)
Age (years)	45.0 [33.0–58.0]
Time from first symptoms to diagnosis of HLH (days)	12.0 [8.0–15.0]
Underlying condition	
Known underlying immunodepression	8 (38%)
Malignancy	5 (24%)
Infection	12 (57%)
Rheumatic disease (AOSD or sJIA)	6 (29%)
Medication	1 (4.8%)
Unknown	1 (4.8%)
Mean number of underlying conditions	2 (1–2)
Fever	21 (100%)
Rash	6 (29%)
Arthralgia/Arthritis	6 (29%)
Hepatosplenomegaly	15 (71%)
Lymphadenopathy	12 (57%)
Serositis	9 (43%)
Neurological symptoms	5 (24%)
Intensive care unit admission	10 (48%)
White blood cells (G/L) (4.0–10.0) *	4.3 [3.1–14.4]
Neutrophils (G/L) (1.8–7.5) *	3.3 [1.9–11.2]
Haemoglobin (g/dL) (13.0–17.0) *	7.8 [7.2–7.9]
Platelets (G/L) (150.0–388.0) *	71.0 [40.0–141.0]
Ferritin (µg/L) (20.0–336.0) *	13,835.0 [6072.5–27,001.0]
Glycosylated ferritin (% of total ferritin)	15.0 [12.5–19.8]
Triglycerides (mmol/L) (0.4–1.7) *	3.5 [3.0–4.5]
Fibrinogen (g/L) (2.0–4.0) *	2.4 [1.5–4.5]
Aspartate aminotransferase (U/L) (15.0–37.0) *	167.0 [58.0–345.0]
Lactate dehydrogenase (U/L) (87.0–241.0) *	635.0 [ 314.0–1263.0]
CRP (mg/dL) (0.0–0.8) *	116.0 [90.4–155.5]
Hemophagocytosis images on bone marrow aspirates	16 (76.2)
H score	244.0 [224.0–267.0]

AOSD: adult-onset Still’s disease; HLH: hemophagocytic lymphohistiocytosis; sJIA: systemic-onset juvenile idiopathic arthritis; * Numbers in parenthesis are the reference values; IQR; interquartile range.

**Table 2 jcm-11-05799-t002:** Patients’ characteristics and outcome.

Patient	Age	Sex	Predisposing and Precipitating Factors	H Score	sHLH Treatment and Line Number	Anakinra Efficacy *	Anakinra Duration (Days)	CTC Duration (Days)	Live Status at Last Visit	Outcome (or Cause of Death)
1	38	M	Tuberculosis, COVID-19	309	1. ANA	Yes	13	0	Alive	Recurrence of IRIS without criterion of sHLH at 1 month.
2	33	M	HIV, tuberculosis, IRIS, CMV and EBV replication	224	1. CTC, 2. ANA	Yes	67	90	Alive	Favourable clinical evolution and good tolerance of antibiotic therapy. Continued tuberculosis treatment during 9 months.
3	73	F	Crohn’s disease treated with azathioprine	299	1. CTC, 2. VP16, 3. IVIG, 4. ANA, 5. HLH-2004	No	13	120	Deceased	Uncontrolled sHLH of unknow origin.
4	66	M	MDS	214	1. CTC, 2. ANA	Yes	20	14	Deceased	sHLH recurrence after decrease of anakinra dosage, good evolution after re-increasing anakinra to 400 mg/d and chemotherapy with azacytidine, but refusal to continue treatment.
5	51	M	HL, EBV replication	262	1. ANA + CTC	Yes	5	7	Alive	Initiation of chemotherapy 7 days after sHLH diagnosis
6	64	M	MDS	218	1. CTC, 2. ANA	Yes	90	150	Deceased	Initial response to ANA, followed by azacytidine initiation with good efficacy. MDS recurrence with Sweet’s syndrome 3 years later.
7	20	M	IJA, tocilizumab	210	1. CTC, 2. ANA	Yes	###	###	Alive	Slow normalization of biological parameters, long-term ANA continuation without recurrence of sHLH
8	17	F	Still’s disease, Parvovirus B19	230	1. ANA + CTC	Yes	180	1420	Alive	ANA efficacy but sHLH recurrence at 1 then at 2 months, controlled with ANA increase and CSA. Recurrent skin eruption under ANA, switch to tocilizumab.
9	41	F	Still’s disease	224	1. CTC, 2. VP16, 3. CSA, 4. VP16, 5. ANA	Yes	660	660	Alive	Two resolved sHLH recurrences after increasing the anakinra dosage to 200 mg/d
10	61	M	Infection (Rickettsia tiphy)	232	1. ANA	Yes	5	0	Alive	Resolution of sHLH under anakinra then treatment with doxycycline 21 days for an infection with *Rickettsia tiphy.* No recurrence.
11	45	F	Still’s disease	279	1. CTC, 2. ANA	Yes	14	610	Alive	Good efficacy of anakinra but pain at the injection site. Switch to tocilizumab with new cutaneous intolerance. Initiation of methotrexate allowing complete withdrawal of CTC.
12	51	M	Graft rejection	317	1. CTC, 2. ANA	Yes	21	130	Alive	Favourable evolution of inflammation and liver disease under anakinra. Discontinuation of anakinra and continuation of corticosteroid alone after 21 days due to erysipelas.
13	35	F	Tuberculosis	294	1. ANA, 2. CTC	Yes	45	180	Alive	Efficacy of anakinra pending liver biopsy results. Then, anti-tuberculosis therapy and CTC.
14	28	F	Kidney transplant, Tuberculosis, IRIS	267	1. CTC, 2. ANA	Yes	120	28	Alive	IRIS resistant to CTC. Dramatic efficacy of anakinra, maintained during the first 4 months of anti-tuberculosis treatment.
15	6	M	JIA, Parvovirus primoinfection	254	1. IVIG, 2. CTC, 3. ANA, 4. CSA + IVIG + CTC	No	210	90	Alive	Failure of IVIG, CTC and then anakinra. sHLH finally suppressed by the association of CTC, CSA and IVIG. Resumption of anakinra as background treatment.
16	4	F	JIA	244	1. ANA + CTC	Yes	60	300	Alive	Good efficacy of ANA after increasing dosage, switch to canakinumab but relapse requiring a new switch to tocilizumab and then to baricitinib.
17	55	F	HL	252	1. ANA	Yes	6	0	Alive	Dramatic response to anakinra allowing diagnostic investigations of the malignant hemopathy without CTC. Initiation of chemotherapy thereafter.
18	58	M	Post-infection (bacteriemia)	187	1. ANA + CTC	Yes	17	7	Alive	sHLH of undetermined aetiology; *Sreptococcus Suis* bacteriemia with arthritis of the right shoulder. Favourable evolution without recurrence.
19	35	F	Infection	225	1. ANA	Yes	8	0	Alive	Cardiogenic shock on stress heart disease and sHLH due to ENT infection. Favourable evolution of sHLH under anakinra alone, no recurrence.
20	58	M	Infection (Knee prothesis) and medication	246	1. IVIG, 2. CTC, 3. ANA	Yes	14	17	Alive	Cardiogenic shock complicating knee prosthesis infection with *Staphylococcus aureus* and *Candida krusei*. Need for ECMO. Favourable evolution under antibiotics/antifungal therapy and ANA.
21	77	M	Infection, urothelial carcinoma	208	1. ANA	Yes	14	0	Alive	Favourable evolution under anakinra as monotherapy of sHLH due to multiple intra-abdominal *Candida* collections, in the setting of an underlying urothelial cancer

ANA: anakinra; CSA: ciclosporine; CTC: corticosteroids; HL: Hodgkin’s lymphoma; HLH-2004: etoposide-dexamethasone-CSA; IRIS: Immune reconstitution inflammatory syndrome; MDS: myelodysplastic syndrome; JIA: juvenile idiopathic arthritis; VP16: etoposide. * As reported by the treating physician.

**Table 3 jcm-11-05799-t003:** Evolution of clinical and biological features under anakinra.

Parameters (*n* *)	Day 0	Day 7	*p*-Value
Fever (*n* = 18)	16 (88.9%)	3 (16.7%)	**0.021**
CRP (*n* = 15)	116	35.1	**<0.001**
Ferritin (µg/L) (*n* = 15)	10,044	2014	**0.011**
Triglycerides (mmol/L) (*n* = 5)	3.7	2.7	1.000
Aspartate aminotransferase (U/L) (*n* = 19)	130	48	**0.025**
Haemoglobin (g/dL) (*n* = 20)	8.9	9.1	0.946
White blood cells (G/L) (*n* = 20)	6.7	5.1	1.000
Platelets (G/L) (*n* = 20)	68	118.5	0.417

* Number of patients with data available at D0 and D7 included in the analysis. Statistically significant *p* values (<0.05) are shown in bold font.

## Data Availability

All the data are provided within the text or as Appendix A.

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
