# Peer review of "Anakinra for the Treatment of Hemophagocytic Lymphohistiocytosis: 21 Cases"

_jcm, 2022, doi:10.3390/jcm11195799_

Round 1
Reviewer 1 Report
The manuscript – Anakinra for the treatment of hemophagocytic lymphohistio-2 cytosis: 21 cases – conduct a comprehensive evaluation for 21 sHLH patients who receive anakinra. They track the evolution of clinical and biological features under anakinra. They showed a dramatical decrease of CRP concentration after 7 days and the reduction of the fever occurrence after ~ 1 day. The side effect of anakinra also seemed to be well tolerated. Their results brought a practical and safe potential treatment for sHLH when using Anakinra.
Although the study showed a significant effect when using Anakinra, a few major and minor should be still taken into consideration. One concern is that there is no decent comparison for sHLH patients without anakinra treatment. As the author collected local database, it would be significantly helpful if they can compare without anakinra treatment or other treatment (e.g., CTC)
L74: H score >169 why? The reference did not have the same and clear criteria.
L106-L109: there are several line therapies, could the author briefly explain in details?
Author Response
Thank you for reviewing our manuscript and for your relevant comments. We will answer point-by-point to your comments.
1.The manuscript – Anakinra for the treatment of hemophagocytic lymphohistio-2 cytosis: 21 cases – conduct a comprehensive evaluation for 21 sHLH patients who receive anakinra. They track the evolution of clinical and biological features under anakinra. They showed a dramatical decrease of CRP concentration after 7 days and the reduction of the fever occurrence after ~ 1 day. The side effect of anakinra also seemed to be well tolerated. Their results brought a practical and safe potential treatment for sHLH when using Anakinra.
R: Thank you for this evaluation of our work.
2.Although the study showed a significant effect when using Anakinra, a few major and minor should be still taken into consideration. One concern is that there is no decent comparison for sHLH patients without anakinra treatment. As the author collected local database, it would be significantly helpful if they can compare without anakinra treatment or other treatment (e.g., CTC).
R: Unfortunately this study was not designed for that, it is a series of cases identified because they had received anakinra. For a comparative study, a prospective collection will be needed (notably because of the many underlying causes and identification/prescription habits in the different services). A clinical trial is currently being set up for this purpose.
3.L74: H score >169 why? The reference did not have the same and clear criteria.
R : Following your comment, in section method, we specify the reference of the publication, in which is noted that « the best cutoff value for HScore was 169, corresponding to a sensitivity of 93%, a specificity of 86%, and accurate classification of 90% of the patients »
4.L106-L109: there are several line therapies, could the author briefly explain in details?
R : As described in our modified manuscript and as you suggested, on page 4, we explain the different treatment lines in more detail.
Anakinra was used as first-line treatment for sHLH in 10 patients. In 6 patients (five with infection, one with Hodgkin’s lymphoma), anakinra was administered as mono-therapy without corticosteroids. None of this six patients required a new line of treat-ment for sHLH; only one patient was switched to oral corticosteroids to facilitate drug intake. In 4 patients (two with rheumatic disease, one with Hodgkin’s lymphoma and one with infection), anakinra was combined with corticosteroids. None of these patients warranted a new line of treatment. Anakinra was used as a second line treatment after failure of corticosteroid therapy alone in 7 patients (two with infection, two with myelodysplastic syndrome, one with graft rejection, two with rheumatic disease). None of this seven patients required a new line of treatment for sHLH. Anakinra was pre-scribed in third line in two cases (one with rheumatic disease, one with infection), after failure of polyvalent immunoglobulin and then steroid therapy alone. In patient 15, who had juvenile idiopathic arthritis and primary parvovirus B19 infection, a fourth line of treatment with ciclosporine was required after failure of anakinra. Anakinra was used as a fourth-line therapy in patient 3 who presented sHLH in the context of Crohn's disease treated with azathioprine, after the use of corticosteroids, polyvalent immuno-globulin and etoposide. Anakinra was not effective in this patient. Finally, anakinra was used as 5th line therapy in patient 9 with Still's disease, without the need for a new line of therapy.
Reviewer 2 Report
The use of anakinra in the treatment of primary and secondary HLH is widely documented in the literature. The work does not add new information in this regard. The study has some important limitations. In this work, the real efficacy of anakinra does not seem to be irrefutable.
1) Anakinra was used as first line therapy in 10 cases , as second or more line in 11 out of 21 patients . The effectiveness of anakinra is difficult to assess when used in combination with other drugs, especially if used in a second or more line of treatment
2) Primary HLH should be genetically investigated and possibly ruled out in patients enrolled in the study. Infections,rheumatological and/ or neoplastic diseases can often be triggers for the unmasking of a primary HLH. On the other hand, the median high levels of ferritin, more than 10,000 mcg / L, could suggest a genetically determined form of HLH in some of these patients
Author Response
Thank you for reviewing our manuscript and for your relevant comments. We will answer point-by-point to your comments.
1. The use of anakinra in the treatment of primary and secondary HLH is widely documented in the literature. The work does not add new information in this regard. The study has some important limitations. In this work, the real efficacy of anakinra does not seem to be irrefutable.
R: The efficacy of anakinra is not well documented in the literature; all information comes from small retrospective series and post-hoc analysis of a trial in sepsis that was not designed to assess the efficacy of anakinra in sHLH and whose criteria for retaining HLH remain debatable. Several authors propose that anakinra should be used, but a randomised trial is widely awaited. Our aim is to report a series of observations. These are consistent with the efficacy of anakinra and add original cases to the literature, which will inform future systematic reviews. Limitations are inherent in a retrospective work which, although original, contains missing data. It is ultimately real life data. The difficulty is even greater for a treatment used outside of approval, in a condition associated with multiple pathologies and different patterns of use. In the end, what unites these observations is 1) the existence of HLH, 2) treatment with anakinra. Only prospective work will provide precise answers, but until such a trial sees the light of day and produces results, it is useful to communicate these data. The limitations of our work have been announced in the text and are reinforced in the discussion section of the revised manuscript, page 8:
Furthermore, judging the efficacy of anakinra in sHLH is made more difficult by the fact that it is a disease that is itself associated with many different pathological conditions that warrant specific treatment, and as anakinra is not currently approved for use in this indication, the administration regimens are left to the discretion of the clinician. However, our data are consistent with the efficacy of anakinra and add original cases of anakinra use in real life to the literature, which will inform future systematic reviews and are useful until a prospective work provide precise answers.
2. Anakinra was used as first line therapy in 10 cases, as second or more line in 11 out of 21 patients . The effectiveness of anakinra is difficult to assess when used in combination with other drugs, especially if used in a second or more line of treatment.
R : As described in our modified manuscript, on page 4, we detail the necessity or not of using a new line of treatment after using anakinra:
Anakinra was used as first-line treatment for sHLH in 10 patients. In 6 patients (five with infection, one with Hodgkin’s lymphoma), anakinra was administered as mono-therapy without corticosteroids. None of this six patients required a new line of treat-ment for sHLH; only one patient was switched to oral corticosteroids to facilitate drug intake. In 4 patients (two with rheumatic disease, one with Hodgkin’s lymphoma and one with infection), anakinra was combined with corticosteroids. None of these patients warranted a new line of treatment. Anakinra was used as a second line treatment after failure of corticosteroid therapy alone in 7 patients (two with infection, two with myelodysplastic syndrome, one with graft rejection, two with rheumatic disease). None of this seven patients required a new line of treatment for sHLH. Anakinra was pre-scribed in third line in two cases (one with rheumatic disease, one with infection), after failure of polyvalent immunoglobulin and then steroid therapy alone. In patient 15, who had juvenile idiopathic arthritis and primary parvovirus B19 infection, a fourth line of treatment with ciclosporine was required after failure of anakinra. Anakinra was used as a fourth-line therapy in patient 3 who presented sHLH in the context of Crohn's disease treated with azathioprine, after the use of corticosteroids, polyvalent immuno-globulin and etoposide. Anakinra was not effective in this patient. Finally, anakinra was used as 5th line therapy in patient 9 with Still's disease, without the need for a new line of therapy.
3. Primary HLH should be genetically investigated and possibly ruled out in patients enrolled in the study. Infections,rheumatological and/ or neoplastic diseases can often be triggers for the unmasking of a primary HLH. On the other hand, the median high levels of ferritin, more than 10,000 mcg / L, could suggest a genetically determined form of HLH in some of these patients
R: Primary HLH occurs mostly in children and consists of repeated episodes. In our series the mean age was 43 years and the age of the youngest patient reported was 4 years. None of the patients had a history of HLH, nor a family history. The ferritin level is not correlated with the cause (although it is known that in Still's disease children have rather less ferritin) and a recent series of extreme hyperferritinemia (Fauter M, Mainbourg S, El Jammal T, Guerber A, Zaepfel S, Henry T, Gerfaud-Valentin M, Sève P, Jamilloux Y. Extreme Hyperferritinemia: Causes and Prognosis. J Clin Med. 2022 Sep 16;11(18):5438. doi: 10.3390/jcm11185438) found only one case of pHLH (NLRC4 GOF, which started in early childhood) out of 495 adult patients with a ferritin level >5000 µg/L. In contrast, 90 patients had secondary HLH. Although it cannot be stated that there is no genetic predisposition, our patients had rather the characteristics of secondary HLH, for which most had an identifiable trigger. In the end, whether the HLH was secondary or not did not alter the conclusion of the study which found a favourable effect of anakinra treatment.
Round 2
Reviewer 2 Report
no further comments